# Single-Port Robot-Assisted Radical Prostatectomy: Where Do We Stand?

Antonio Franco [1,2], Antony A. Pellegrino [3,4], Cosimo De Nunzio [2], Morgan Salkowski [1], Jamal C. Jackson [1], Lucas B. Zukowski [1], Enrico Checcucci [5], Srinivas Vourganti [1], Alexander K. Chow [1], Francesco Porpiglia [6], Jihad Kaouk [7], Simone Crivellaro [3] and Riccardo Autorino [1,*]

[1] Department of Urology, Rush University, Chicago, IL 60612, USA
[2] Department of Urology, Sant'Andrea Hospital, La Sapienza University, 00189 Rome, Italy
[3] Department of Urology, University of Illinois at Chicago, Chicago, IL 60612, USA
[4] Unit of Urology/Division of Oncology, IRCCS San Raffaele Scientific Institute, 20132 Milan, Italy
[5] Department of Surgery, Candiolo Cancer Institute, FPO-IRCCS, Candiolo, 10060 Turin, Italy
[6] Division of Urology, San Luigi Gonzaga Hospital, University of Turin, Orbassano, 10043 Turin, Italy
[7] Department of Urology, Cleveland Clinic, Cleveland, OH 44106, USA
[*] Correspondence: ricautor@gmail.com

**Abstract:** In 2018, the da Vinci Single Port (SP) robotic system was approved by the US Food and Drug Administration for urologic procedures. Available studies for the application of SP to prostate cancer surgery are limited. The aim of our study is to summarize the current evidence on the techniques and outcomes of SP robot-assisted radical prostatectomy (SP-RARLP) procedures. A narrative review of the literature was performed in January 2023. Preliminary results suggest that SP-RALP is safe and feasible, and it can offer comparable outcomes to the standard multiport RALP. Extraperitoneal and transvesical SP-RALP appear to be the two most promising approaches, as they offer decreased invasiveness, potentially shorter length of stay, and better pain control. Long-term, high-quality data are missing and further validation with prospective studies across different sites is required.

**Keywords:** single port; radical prostatectomy; robotic surgery; extraperitoneal

## 1. Introduction

Radical prostatectomy is the preferred therapeutic option in men with localized prostate cancer [1]. Since the US Food and Drug Administration (FDA) approval for da Vinci-assisted prostate surgery in 2001, robotic prostatectomy has become the most commonly performed robotic oncologic procedure in the United States [2].

By 2013, 85% of all radical prostatectomies were performed robotically, highlighting the rapid rate of diffusion of the surgical robot in urology [3]. As a matter of fact, the use of robotic assistance in laparoscopic surgeries has exhibited minor length of hospital stay and reduced perioperative blood loss [4]. Furthermore, questionable and controversial studies have shown that robotic radical prostatectomy may have improved erectile performance postoperatively, as well as delivering superior rates of urinary continence when compared to other approaches [5,6].

Subsequent generations of the da Vinci robotic platform have been released over the past 20 years, sharing a multi-arm design with a fixed laparoscopic camera. Laparoendoscopic single-site surgery (LESS) and natural orifice transluminal endoscopic surgery (NOTES) were meant to become the future of minimally invasive surgery, with the aim of minimizing postoperative pain and incision-related complications [7]. However, the rigid instrumentation, extensively long surgical time, significant challenge with instrument clashing, and limited operative space were all important factors which limited the widespread adoption of these single-site techniques, despite their potential advantages of reducing surgical morbidity, improving cosmesis, and reducing pain [8].

In 2014, the initial clinical series of single-port (SP) robotic surgery using a purpose-built robotic platform were reported [9]. A few years later, in 2018, the da Vinci Single Port (SP) system was approved by the FDA for urologic surgery. Instead of the multiple trocars commonly used during standard multiport robotic procedures, this novel platform accommodates all the robotic instruments and camera through a single multichannel 2.5 cm port inserted through a single skin incision. Robotic-assisted radical prostatectomy (RALP) is the preferred surgical treatment option for prostate cancer [2], and it has been traditionally performed using a transperitoneal approach. With the implementation of the SP robotic system, alternative approaches have been explored with the aim of maximizing the benefits this new platform can offer [10]. Since the initial description of SP-RALP by Kaouk et al. [11], many centers have reported different techniques and outcomes with this novel procedure.

The aim of the present review is to describe the different approaches and techniques for SP-RALP as well as to summarize reported outcomes to date.

## 2. Literature Search

An electronic search of MEDLINE using a combination of MeSH terms and free text from introduction of the da Vinci SP platform from 2018 until December 2022 was performed. Research terms used for the research were the following: "Single Port Robotic Radical Prostatectomy" or "Single Port Prostatectomy" or "Single Port Robot Prostatectomy". All the references of key reviews on SP Radical Prostatectomy were also screened. Only English language articles were included. Titles and abstracts were analyzed. After this initial screening, a full-text review was conducted to confirm the selected articles' eligibility for inclusion. Editorials, commentaries, abstracts, and book chapters were excluded from the analysis.

## 3. Evidence Synthesis

### 3.1. The SP System

The da Vinci SP (Intuitive Surgical, Sunnyvale, CA, USA) is a novel surgical system with multiple innovations. It uses a single port of 27 mm that allows the introduction of an 8 mm articulating flexible camera and three articulating 6 mm instruments. The 25 mm cannula can be placed directly within the 27 mm incision, or within a GelPOINT® advanced access platform. Although it shares some common features with its predecessors—such as Endowrist manipulators, three-dimensional visualization with magnification and scaled movement, and tremor reduction—the single-port system introduces new peculiarities: a flexible camera, which can rotate in all directions and therefore different perspective angles can be achieved while the instruments maintain a fixed position. Furthermore, a specific feature of the SP called "relocation" allows the entire platform to be moved in any direction around its fulcrum. The camera and each instrument are positioned in the 12, 3, 6, and 9 o'clock positions within the trocar. Port placement is flexible and allows for 360° of robotic docking. Furthermore, a new visual icon, termed the 'Navigator', has been introduced, improving cooperation between the instruments and the camera. In doing so, all the instruments can be tracked simultaneously. Finally, the 'Cobra Mode' feature, essentially a centered and ~30° flexed position of the camera with optimal instruments visualization, supports the surgeon in identifying the ideal position of the camera within the various instruments during each surgical step.

These new instrument mechanics and innovative features create several specific technical adaptations from the multiport approach, including minimizing instrument clashing and maximizing workspace within the patient, additional movements, and changing angulation. On the other hand, the surgical field is reduced, as well as rotation of the instruments, thus requiring expertise and incremented coordination by the surgeon [12].

Different approaches have been described for SP-RALP, each one with advantages and disadvantages (Table 1).

**Table 1.** Different SP approaches for RALP procedure: advantages and disadvantages.

| Approach | Incision | Advantages | Disadvantages |
|---|---|---|---|
| Transperitoneal | Above umbilicus | Wide operative space—Faster learning curve—Easy access to lymph node regions | Need for Trendelenburg—More challenging in patients with previous abdominal surgery—Risk of bowel injury—Postoperative ileus |
| Extraperitoneal | Suprapubic | Supine position—Avoidance of peritoneal cavity (especially in patients with difficult abdomen)—No ileus—Lower need for narcotics—Faster recovery/hospitalization | Reduced working space—Initial learning curve—Risk of increased $CO_2$ absorption |
| Perineal | Perineal Semilunar | Lower incidence postop—Strictures—Avoidance of peritoneal cavity in patients with difficult abdomen—Preservation of anterior structures in the Retzius space | Higher risk of rectal injury—Difficult nerve sparing—Reversed anatomy (steep learning curve) |
| Transvesical | Suprapubic | Supine position—Avoids abdominal cavity—No need for bladder mobilization | Limited working space in case of large prostates—Cannot be used in case of bladder cancer—Allows only limited lymph nodes dissection—Risk of increased $CO_2$ absorption |
| Retzius-sparing | Above umbilicus | Supine position—Preservation of anterior structures in the Retzius space—Improved early continence | Small workspace—No lateral aiming point when dissecting the lateral pedicles—Inverted anatomical relationship between the bladder and prostate during dissection and reconstruction—Steep learning curve—Higher rates of PSM in pT3 tumor |

*3.2. Transperitoneal Approach*

As transperitoneal approach represented the most familiar approach for multiport RALP, it is not surprising that it was also the one initially preferred with the adoption of the SP platform. Here, the patient is placed in a 25° Trendelenburg position, an incision is made above the umbilicus, and the peritoneum is entered under direct vision via Hasson technique. An Alexis wound retractor is introduced through the incision, and a GelPOINT Access Platform is secured to the retractor. A valveless Airseal® port can be either placed through a different fascial incision (using same skin incision) or through the GelPOINT retractor. The SP robot is then docked to the SP access port, and the robotic instruments are introduced. The dissection is then handled in a similar way to a multiport transperitoneal RARP, with either a posterior or anterior approach.

*3.3. Extraperitoneal Approach*

The extraperitoneal approach for the multiport system has been described [13,14]. Albeit feasible, it is limited by a restricted working space as well as instrument clashing, which did not allow a wide adoption of this approach. The SP platform allows to overcome these limitations, making it more feasible and potentially more appealing [15]. In this case, there is no need for the Trendelenburg position as the patient can lay supine, with a remarkable advantage in terms of anesthesiology support. A single, horizontal infraumbilical incision is made; blunt finger dissection is used to develop the Retzius space to the pubic bone. Recently, a novel SP access kit has been developed, which consists of a wound retractor, an inflatable plastic sphere which provides extra space for the robotic arms—working as a "floating" platform—and an SP robotic trocar. Once the Retzius space is entered, the procedure is carried out in a similar way to the transperitoneal approach.

### 3.4. Extraperitoneal versus Transperitoneal Approach

As of the date of publication, there are only two studies comparing extraperitoneal to transperitoneal SP-RALP. The first experience was described by Kaouk et al. [16], demonstrating a significantly shorter postoperative hospital stay and decreased need for postoperative narcotics, as well as shorter operative time for the extraperitoneal cohort. Later, the first and largest multi-institutional propensity score-matched study comparing the two approaches was reported [17] (Table 2). Results largely echoed findings from the previous study, with exception of the operative time, which was longer in the extraperitoneal group (median 206 vs. 155 min, $p < 0.001$). The authors justified this finding by the different surgeon experience, additional operative time required for creating the extraperitoneal space, and more frequent lymph node dissection cases in the extraperitoneal cohort [17].

**Table 2.** Outcomes of transperitoneal versus extraperitoneal SP RALP (adapted from Abou Zeinab et al. [17]).

| Outcome | TP (*n* = 238) | EP (*n* = 238) | *p* Value |
|---|---|---|---|
| Operative Time, min | 206 | 155 | <0.001 |
| Estimated blood loss, mL | 150 | 75 | <0.001 |
| Length of hospitalization, h | 7.5 | 14 | <0.001 |
| Pain scale at discharge | 2 | 2 | 0.923 |
| Postoperative complications, *n* (%) | 39 (16.4%) | 32 (13.4%) | 0.368 |
| Positive Surgical Margin, *n* (%) | 55 (23.3%) | 61 (26.9%) | 0.376 |
| Continence rate @6 Months, *n* (%) | 125 (86.8%) | 163 (87.2%) | 0.923 |

TP = Transperitoneal; EP = Extraperitoneal; Values expressed as median for continuous variables.

In summary, the extraperitoneal approach appears to be a less invasive approach, consequently resulting in less operative time and fewer days of hospitalization, with possible same-day discharge (SDD) in most cases. The possibility of avoiding the peritoneum, thus avoiding peritoneal irritation and postoperative ileus, also allows to perform cases with extensive previous abdominal surgery, and to minimize the postoperative use of narcotics. Furthermore, lack of steep Trendelenburg and pneumoperitoneum may also expedite postoperative recovery and facilitate anesthesia. All these factors may be responsible for shortening hospitalization length [13,18]. On the other hand, the extraperitoneal approach is potentially associated with increased $CO_2$ absorption, resulting in hypercapnia and, possibly, systemic acidosis [19]. Although these complications appear to be rare, surgeons must be aware of this problem, and pneumo pressure should be kept at lower levels compared to what is usually used in transperitoneal cases. A history of prior laparoscopic extraperitoneal mesh herniorrhaphy or kidney transplantation might represent another relative contraindication to the extraperitoneal approach, as access to the retropubic space would be limited due to adhesions and scar fibrosis. In this case, the transperitoneal approach may be more feasible [13]. The learning curve for extraperitoneal radical prostatectomy (EPRP) might be a steep one due to restricted working space, therefore young or less-experienced surgeons may prefer starting with the transperitoneal approach to achieve more confidence and dexterity with the single-port platform. Regarding oncological and functional outcomes, there appear to be no significant differences between the two approaches. Positive surgical margins were comparable, and the stress incontinence rate was similar at 3 and 6 months [14,20].

### 3.5. Perineal Approach

First described by Young in 1905, perineal radical prostatectomy was the most common access for surgical treatment of prostate cancer for almost seven decades. However, this technique became less favored due to technical complexity and the narrow operative space.

Robotic-assisted perineal radical prostatectomy was shown to be feasible, despite some technical challenges. However, limited clinical series exist to date [21].

The Cleveland Clinic group recently reported the only clinical series on SP robotic perineal radical prostatectomy [22]. Briefly, the patient is placed in the lithotomy position and a 4–5 cm perineal incision is made. After dissecting the subcutaneous tissue and dividing the central tendon, the external sphincter muscle is retracted superiorly. Then, the GelPOINT device is placed, and the robot is docked. The posterior prostatic space is developed, levator ani fibers are split along the lateral aspects of the prostate, and the Denonvilliers fascia is opened to find the plane of seminal vesicles and vasa deferens; bilateral vascular pedicles are identified and ligated followed by prostatic apex and urethra dissection; the bladder neck is identified and opened and vesicourethral anastomosis is performed according to the habitual technique. When lymphadenectomy is indicated, the access to bilateral pelvic lymph nodes is achieved with the same incision and does not require another access as previously described. The inferior lateral perivesical space, that was initially prepared after the splitting of the levator ani muscle, is now utilized for gaining exposure to the obturator fossa and the inferior edge of the external iliac vein. To note, the anatomy is reversed from the typical retropubic approach: the obturator structures will be encountered before the external iliac vein. Kaouk et al. compared SP transperineal radical prostatectomy to standard multiport transperitoneal RALP performed by the same surgeon at the beginning of the SP experience. Overall, the study showed equivalent functional and oncological outcomes at 12 months, but a higher complication rate and a higher positive surgical margin detection was recorded in the SP group (38.5% vs. 7.7%, $p < 0.01$) [22].

In conclusion, robotic SP perineal radical prostatectomy is a feasible but challenging procedure. Its role is limited to very selected cases and in centers with enough expertise to perform this procedure.

### 3.6. Transvescical Approach

After describing the single-port transvescical approach for simple prostatectomy [23,24], Kaouk et al. reported an initial clinical experience for SP transvescical radical prostatectomy. The oncological and functional outcomes were comparable with other approaches, although the sample size was limited [25]. In this procedure, the patient is placed in a supine position and an incision is made 4 cm above the pubic symphysis; after distension of the bladder, a GelPOINT trocar is percutaneously inserted into the bladder and the multichannel SP cannula is inserted through the GelPOINT GelSeal cap. The pneumo-vesicum is then created with carbon dioxide insufflation and is robot docked. Access to the prostate is gained by incising the bladder neck distal to the trigone and ureteral orifices, thus enabling immediate clear visualization of the peripheral zone of the prostate. The dissection continues toward the apex and then the plane of seminal vesicles is identified; the infratrigonal intravesical incision is then extended circumferentially to complete the bladder neck dissection; the anterior plane of the prostate is then prepared, and anastomosis completed. The benefits of this approach include avoiding unnecessary dissection and mobilization of bladder, bowel mobilization, any need for lysis of adhesions in patients with previous surgery, and Trendelenburg positioning [25]. Moreover, $CO_2$ is minimally absorbed; consequently, patients with significant cardiopulmonary comorbidities may profit an epidural anesthesia rather than general anesthesia. On the other hand, limitations of this approach are mostly related to bladder diseases, such as diverticula, trabeculation, and augmented bladder capacity. Above all, a large volume of the prostate might render this procedure more complex [25].

### 3.7. Retzius-Sparing Approach

A Retzius-sparing approach was first described by Bocciardi's group in Milan [26]. The main goal of this technique is to leave the bladder in its native anatomical position, sparing Santorini's plexus, endopelvic fascia, puboprostatic ligaments, and the other anterior compartment structures that in robotic radical prostatectomy have been associated with improved urinary continence rates compared with anterior approaches [27–29]. In fact,

early continence upon catheter removal has been reported in up to 92% of patients [30]. Patient positioning is the same as extra or transperitoneal approach. A 2.5 cm vertical incision is made superior to the umbilicus with the peritoneum. The GelPOINT® Advanced Access Platform is then assembled. Prostatectomy is then performed as described previously by Galfano et al. [26]. Several notable modifications of surgical technique are facilitated by the SP platform. Firstly, the bladder is lifted by the Cadiere forceps at the 12 o'clock position to develop the interfascial or intrafascial plane between the prostatic and Denonvilliers' fascia. "Cobra" mode camera allows to reach the apices without requiring suspension sutures through the abdominal wall as described in the multiport technique. The dissection of the ventral aspect of the gland is facilitated by the increased degrees of camera articulation provided by the flexible scope.

SP Retzius-sparing radical prostatectomy (SP-rsRALP) has been described in the past three years by initial experience with the cadaveric model and afterward with few series of patients [31–34]. The largest cohort of SP-rsRALP was presented by Balasubramanian S. et al. who compared this approach to extraperitoneal and transperitoneal ones. The three SP-RALP approaches appear to be safe and feasible, with similarity in terms of perioperative outcomes, oncologic outcomes, and postoperative pain control. Faster and improved returns of both continence and erection were associated with this technique [31]. However, this surgical procedure presents a steep learning curve and potential complications when compared to other accesses [33]. Moreover, working in a smaller operative space makes SP-rsRALP on larger glands technically challenging according to several reports [35]. In fact, other studies reported that rsRALP offers a higher positive surgical margin (53% rate) than classic RALP, especially for anterior tumors [36]. Other limitations may include no lateral aiming point when dissecting the lateral pedicles of the prostate, a poor vision of the bladder neck during dissection and consequently of the position of the ureteric orifices.

## 4. Oncological and Functional Outcomes

Even if the adoption of the SP robotic system is still at its initial stages, SP surgery is rapidly gaining popularity worldwide and becoming appealing to the eyes of expert robotic surgeons. While high-quality comparative studies are lacking, early studies have supported acceptable perioperative outcomes, comparable to the ones of traditional multi-port robotic surgery.

Regarding SP-RALP, more than 30 series have been published during the last three years, with almost 1000 patients enrolled. In general, the procedure was shown to be feasible and safe. In a systematic review on the subject [20], Checcucci et al. considered 6 studies, including 107 patients. Overall, operating time, estimated blood loss, hospitalization time, and catheterization time were 190.55 min, 198.4 mL, 1.86 days, and 8.21 days, respectively. Oncological outcomes showed a pooled mean number of lymph nodes removed around 8.33, and pooled positive surgical margin rate of 33%. In terms of functional outcomes, pooled continence and potency rates at 12 weeks were 55% and 42%, respectively. Only 15% of minor complications were observed and one major complication overall [20]. Despite the limitations of the extremely short follow-up and the low sample size, the authors stated that functional and oncological outcomes seemed to be promising. However, different surgical approaches among surgeons might have represented a major source of bias. Notwithstanding these limitations, results are in line with previously reported outcomes for multi-port radical prostatectomy (MP RALP) [37–39].

More recently, comparative outcomes of SP versus MP RALP have been assessed in three different systematic reviews (Table 3). Gonzalez et al. [40], Fahmy et al. [41], and Li et al. [42] analyzed 1068, 666, and 1239 patients, respectively. Only Li et al. performed a subgroup analysis based on the different surgical approaches; in fact, they include perineal access, whereas the other two studies took in account only more traditional (transperitoneal and extraperitoneal) approaches [42]. Similar results were observed in terms of operative time, blood loss, continence and potency rates, complication rate, positive surgical margin, and biochemical recurrence. On the contrary, all three studies reported a shorter hospital

stay and a lower requirement for opioids for the SP cohorts [40–42]. Furthermore, cosmetic outcome was certainly addressed as a major advantage for SP surgery, especially for some types of patients [43]. Regarding catheterization time, only Li et al. had investigated this variable: the SP-RALP group demonstrated less catheterization time compared to MP RALP (WMD-1.51 days, $p$ = 0.007) [42]. Again, a shorter catheterization time was associated with reduced invasiveness, whereas most patients who underwent SP-RALP or MP RALP experienced from 5 to 7 days catheterization time postoperatively, as reported in previous series [44]. Only one study performed a cost analysis, reporting higher costs for SP RALP. However, the lower cost of shorter hospitalization may counterbalance the higher cost of surgical consumables, hence the costs for SP-RALP and MP RALP prostatectomy may be comparable [45]. Several studies suggest a major role of SP-RALP in an outpatient setting [15–17]. Certainly, a lower administration of analgesics and opioids will positively also affect hospital stay and facilitate outpatient procedures. Concerning the learning curve of SP surgery, it was observed that shifting from MP to SP procedures can be quickly achieved; however, it must be noted that all the studies included in these analyses are from high-volume centers, where conventional RALP had been well established for many years and performed by expert surgeons.

**Table 3.** Perioperative, functional, and oncological outcomes of SP vs. MP RALP according to published systematic reviews [40–42].

| Ref. | No of Cases | OT, min | EBL, mL | LOS, h | Postop. Complications | PSM | Continence Rate * |
|---|---|---|---|---|---|---|---|
| Li 2022 [42] | 372 SP vs. 815 MP | −4.63 | −11.71 | −17.86 | 1.4 | 0.91 | 1.17 |
| Fahmy 2021 [41] | 298 SP vs. 368 MP | 0.22 | 0.21 | −1.06 | 1.32 | 0.78 | 1.47 |
| Gonzalez 2021 [40] | 324 SP vs. 744 MP | 3.93 | −15.77 | −0.94 | 1.29 | 0.78 | 1.29 |

SP = Single Port, MP = Multi-port; OT = Operative time, EBL = Estimated blood loss, LOS = Length of stay, PSM = Positive surgical margin; * Continence rate at 3 months. Values expressed in mean difference for continuous variables or odds ratio for categorical variables.

In summary, few comparative studies exist, supporting non-inferior peri-operative, functional, and oncological outcomes, a shortened length of stay, and reduced required pain therapy for SP-RALP. Short follow-up, possible selection bias from either strict inclusion criteria, or different approaches or techniques chosen—such as the use of auxiliary ports during procedures or disparate surgical experience between centers—are all factors which limit the comparison between the platforms in terms of functional or oncologic outcomes. In fact, currently available articles on the SP console are mostly represented by limited leaders of the field whose results and experience might not be representative of the average urologic surgeon's capabilities. In addition, some of the included studies feature initial surgeon experience with the platform, creating further biases analysis from unequal experience and familiarity. However, the unique aspects of the novel purpose-built single-port platform could lead to several advantages with respect to the multiarm robotic system. As already mentioned above, the initial investigation showed promising results in furthering the field of single-site/single-port surgery by offering the preservation of triangulation, the EndoWrist technology, and the strength of traction via a single keyhole incision. Single-port radical prostatectomy can be considered a future step for only expert surgeons who have already completed their learning curve for multiple-port robotic prostatectomy. Randomized trials with long-term follow-up are necessary for valid investigation of the real benefits of the single-port system.

Future innovations both in terms of new applicability and platforms are ongoing. For example, SP partial prostatectomy has been described as an alternative to focal therapy for the management of localized low- and intermediate-risk prostate cancer [46]. Natural orifice transluminal endoscopic surgery (NOTES) was conceived as the evolution of the

robotic platform, a completely "scarless" surgery [47], but for this to become feasible, a dedicated robotic platform is needed.

## 5. Conclusions

The da Vinci SP platform represents the latest evolution in technology used in robotic-assisted surgery. In the USA, this platform is rapidly gaining popularity within robotic centers of excellence. Despite the current limited evidence, many expert surgeons have embraced the SP-RALP procedure with encouraging outcomes. The RALP procedure using the SP robotic system has proven to be safe and feasible in expert hands and allows to minimize surgical invasiveness by avoiding the need to access the peritoneal cavity. While high-quality comparative studies are lacking, preliminary experiences from high-volume centers demonstrate promising results in terms of oncological, functional, and perioperative outcomes.

**Author Contributions:** Conceptualization A.F. and R.A.; methodology, A.A.P.; software, S.C.; validation, J.C.J., M.S. and L.B.Z.; formal analysis, A.F.; investigation, S.V. and A.K.C.; resources, J.K.; data curation, C.D.N.; writing—original draft preparation, A.F.; writing—review and editing, A.F.; visualization, E.C.; supervision, F.P.; project administration, R.A.; funding acquisition, none. All authors have read and agreed to the published version of the manuscript.

**Funding:** This research received no external funding.

**Conflicts of Interest:** The authors declare no conflict of interest.

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
