# Peer review of "Single-Port Robot-Assisted Radical Prostatectomy: Where Do We Stand?"

_curroncol, doi:10.3390/curroncol30040328_

Round 1

Reviewer 1 Report

This manuscript is about different approaches and techniques for single port radical prostatectomy (SP-RALP) and summarises the outcomes.

The authors did a great job of illustrating the advantages and disadvantages of the different approaches and methods.

The disadvantages of the different methods and approaches will be more apparent to the readers, like talking about the side effects and their common and the percentage of these side effects. They have to write in the manuscript about the

Otherwise, the paper is well written.

I advise publishing this manuscript after the correction of these little notes and after the correction of some grammatical failures.

Author Response

Thanks for your comment.

We revised all the grammar and spelling mistakes. Advantages and disadvantages and side effects of each approach are highlighted.

Please see the attachment below with all the  highlighted comments.

Reviewer 2 Report

This study was reported the utility of robot-assisted radical prostatectomy using the SP system. Generally, this paper is well written. The reviewer thinks that this paper has useful information for readers. However, the reviewer would like to suggest some critiques to make this paper as follows.

Major revision

1.     On line 20, "SP robotic assisted radical prostatectomy (SP-RARP)" sounds a little strange. “robot-assisted radical prostatectomy using the SP robotic system (SP-RARLP)” is better.

2.     The authors refer to "RALP" in lines 20, 36, etc., but in line 140, they refer to "RARP". The authors should be consistent in their descriptions.

3.     On line 22, what is “counterpart”?

4.     On line 95, what is “minor days”? What is “possible same days”?

5.     On line 109, what is “ERPR”?

6.     On line 218, what is “MR RALP”?

Author Response

Thanks for your remarks.

We revised all the grammar and spelling mistakes, addressing all the above points.

Please see the attachment below with all the highlighted comments.

Reviewer 3 Report

This narrative review gives the reader an overview of current knowledge about single-port robotically assisted radical prostatectomy. To date, although it is great innovation, knowledge is limited regarding this subject. Hence, a systematic review of higher quality would not be possible. Nevertheless, the authors adequately present all available information. 

The paper can be accepted for publication. I understand that the authors are pioneers in single port radical prostatectomy, but self-citations should be significntly reduced .. 

Table 1: The common positive surgical margins in T3 disease is an additional limitation of the Retzious-sparing technique and should be added.   

Author Response

1. This narrative review gives the reader an overview of current knowledge about single-port robotically assisted radical prostatectomy. To date, although it is great innovation, knowledge is limited regarding this subject. Hence, a systematic review of higher quality would not be possible. Nevertheless, the authors adequately present all available information. The paper can be accepted for publication. I understand that the authors are pioneers in single port radical prostatectomy, but self-citations should be significantly reduced.

REPLY: Thank you for your comment. Following your suggestion, we significantly reduced the self-citation. However, please note that most of the literature on this topic has been reported by some of the co- authors of the present systematic review.

2. Table 1: The common positive surgical margins in T3 disease is an additional limitation of the Retzious-sparing technique and should be added.

REPLY: We added the Retzius-sparing technique's limitation about PSM in Table 1.
